# Physicochemical Features and Volatile Organic Compounds of Horse Loin Subjected to Sous-Vide Cooking

**DOI:** 10.3390/foods13020280

**Published:** 2024-01-16

**Authors:** Joko Sujiwo, Sangrok Lee, Dongwook Kim, Hee-Jeong Lee, Soomin Oh, Yousung Jung, Aera Jang

**Affiliations:** Department of Applied Animal Science, Kangwon National University, Chuncheon 24341, Republic of Korea; jokosujiwo@ugm.ac.id (J.S.); free5765@naver.com (S.L.); donguk8282@naver.com (D.K.); doob2029@naver.com (H.-J.L.); osm808@naver.com (S.O.); dbtjd97@naver.com (Y.J.)

**Keywords:** aroma, cooking temperature, cooking time, flavor, tenderness, shear force, volatile organic compound

## Abstract

The purpose of this study was to evaluate the effect of temperature and time of sous-vide cooking method on the characteristics of Thoroughbred horse loin. Sliced portions (200 ± 50 g) were cooked by boiling (control) and sous-vide (65 and 70 °C for 12, 18, and 24 h). The samples were analyzed for proximate composition, pH, color, texture, microstructure, sodium dodecyl sulfate–polyacrylamide gel electrophoresis (SDS-PAGE), microbiology, volatile organic compounds (VOCs), nucleotide content, and fatty acids composition. The color analysis showed decreased redness at elevated temperatures. Improved tenderness, demonstrated by reduced shear force values (36.36 N at 65 °C for 24 h and 35.70 N at 70 °C for 24 h). The micrographs indicated dense fiber arrangements at 70 °C. The SDS-PAGE revealed muscle protein degradation with extended sous-vide cooking. The VOC analysis identified specific compounds, potentially distinctive markers for sous-vide cooking of horse meat including 1-octen-3-ol, decanal, n-caproic acid vinyl ester, cyclotetrasiloxane, octamethyl, and 3,3-dimethyl-1,2-epoxybutane. This study highlights the cooking time’s primary role in sous vide-cooked horse meat tenderness and proposes specific VOCs as potential markers. Further research should explore the exclusivity of these VOCs to sous-vide cooking.

## 1. Introduction

Horse meat is less popular than other meats, such as poultry, pork, beef, and mutton, and accounts for only 0.25% of the total global meat production [1]. Horse meat is unique and has a global consumer base. It is characterized by lower cholesterol and fat content than other meats, with approximately 20% less fat than pork, beef, or poultry [2]. Additionally, owing to its high iron concentration and unsaturated fat content (over 55%), horse meat consumption may offer health benefits to humans [2]. It is known for its distinct flavor profile, which is lean with a slightly sweet aftertaste. Uncooked horse meat has dense fibers because it is typically obtained from retired horses, slaughtered riding horses, or racing horses. Although this meat can be tough, it is rich in minerals and protein and low in fat because of the horse’s unique diet [3]. These characteristics make horse meat an intriguing subject for culinary experimentation, especially when using advanced cooking methods, such as sous vide.

The utilization of thoroughbred horses extends across diverse domains, such as horse sports, leisure activities, and tourism [4]. Concurrently, the unwanted horses are subjected to slaughter for meat consumption. Consequently, the inclusion of thoroughbred horses in this study is imperative, as it serves as a representative sample reflecting the broader horse industry.

Sous-vide cooking is the process of cooking food at specific temperatures and durations within a thermostabilized vacuum-sealed bag. This method involves packaging the product in plastic bags with low oxygen permeability and high-temperature resistance and then cooking the products for extended periods in a thermally stable container at a constant temperature; afterward, the products are stored in a low-temperature condition or served immediately [5]. Vacuum-sealed low-temperature cooking methods have significant appeal to the food industry. This appeal is primarily due to the positive effect of the sous-vide cooking method. The sous-vide method, applied to beef, pork, and poultry, offers advantages such as increased process yield, leading to enhanced juiciness and improved flavor, aroma, color, tenderness, and microbial control, while minimizing nutrient loss [6]. Furthermore, vacuum technology has potential applications not only in the food service sector, which includes restaurants, catering, and food retail, but also in various industries, such as transportation (aviation, railway, and maritime transport), medical facilities (hospitals), healthy food markets, educational facilities (schools), and the military.

Conventional cooking methods differ from the sous-vide method recommended by experts for specific meat types and cuts. Typically, experts advise using temperatures ranging from 58 to 63 °C for 10–48 h for pork, while the conventional cooking of pork often involves temperatures of 75–80 °C [7]. Previous studies have used temperatures span from 60 to 80 °C for 6–24 h on lamb [8] and from 45 to 65 °C for 4 h on beef [9]. Sous-vide cooking enhances the tenderness and juiciness in various cuts of meat, including pork, lamb, and beef [10]. The tenderness achieved through sous-vide cooking is primarily a result of protein denaturation at low temperatures, which weakens the connective tissue by solubilizing collagen and retaining water [11,12,13]. Vacuum-sealed food efficiently conducts heat and prevents lipid oxidation, loss of volatile compounds, and moisture loss, thereby improving food flavor during sous-vide cooking [14].

Previous studies have shown that sous-vide cooking improves the quality of beef [15], lamb [8], poultry [16] and pork [17]. However, there is limited scientific information regarding the effects of sous-vide cooking with variations in duration and temperature on horse meat. Therefore, the objective of this study was to investigate the effect of the sous-vide cooking temperature–time combination on the physicochemical and microbiological characteristics of horse loin.

## 2. Materials and Methods

### 2.1. Preparation of Sample and Experimental Design

The loins (*M. longissimus thoracis et lumborum*) at the thoracic part of three Thoroughbred horses were obtained within seven days of slaughter from a specialized horse meat producer in Republic of Korea (Cheonma Mall, Cheonma Agricultural Company Co., Ltd., Cheonan, Republic of Korea). The Thoroughbred breed was chosen because it is a frequently used breed in the broad horse industry. In this study, three horses were randomly selected without consideration of their sex and were slaughtered at 48 months of age. The meat was then transported to the laboratory under chilled conditions (4 ± 2 °C). Upon arrival, the meat samples were divided into segments weighing 50–200 g each, with a uniform thickness of 2.5 cm. Subsequently, these segments were hermetically vacuum-sealed in food-grade bisphenol A-free vinyl plastic pouches (0.08 mm thick, Daegu, Republic of Korea), using a chamber vacuum packer (Sambo Tech, Lovero SBV-400 TS, Gimpo, Republic of Korea), and promptly frozen. The samples were frozen and stored at −20 °C until further analysis. The cooking duration and temperatures were selected based on initial trials and published research data [10,18,19]. Meat samples from the three horses were evenly distributed among six different treatments. Six different combinations of time (12, 18, and 24 h) and temperature (65 and 70 °C) were used for sous-vide cooking 36 horse loins (*n* = 6 for each treatment). The samples were subjected to sous-vide cooking at 65 °C and 70 °C for 12, 18, and 24 h. A water bath equipped with a temperature sensor and an immersion instrument was used for the sous-vide treatment (LKLabKorea Inc., LB-WD522, Namyangju, Republic of Korea). For the control group, meat was placed in vinyl plastic pouches and cooked in a water bath, with the temperature set to 75 °C, for 40 min, until the core meat temperature reached 72 °C. Prior to sous-vide cooking, the samples, including the control group, were thawed for 30 min in tap water. In addition, the raw sample of horse meat was used for the analysis of microbiology, microstructure, myofibrillar protein, and volatile compounds. After cooking, the samples were subjected to various analyses, including physicochemical assessments (proximate composition, pH, and instrumental color), textural analyses (shear force, myofibril fragmentation index (MFI), and texture profile analysis (TPA)), microstructure evaluation, microbiological analysis, and assessments of flavor characteristics (volatile organic compounds (VOCs), nucleic acid, and fatty acid analyses). To ensure consistency and preserve the sample quality, the cooked samples were rapidly cooled on ice for 1 h.

### 2.2. Proximate Analysis and Cooking Loss

The moisture, crude fat, ash, and crude protein contents were measured according to the Association of Official Agricultural Chemists method [20]. Moisture content was determined using the atmospheric-pressure oven-drying method at 105 °C. The crude fat was extracted using the Soxhlet extraction method with ether. The ash content was determined using the dry ashing method at 550 °C. The crude protein content was determined using the Kjeldahl method. The calculation of crude protein consisted of multiplying the nitrogen concentration by a conversion factor of 6.25. The cooking loss was quantified by measuring the difference in meat weight before and after treatment.

### 2.3. Determination of pH Value

After the thermal treatment, each sample was grounded using a meat grinder. The grinding process was conducted in the chilling room, at a temperature of 4 ± 2 °C, to minimize the changes in sample quality properties. A total of 3 g of the ground sample and 27 mL of distilled water were homogenized, using a PolyTron^®^ PT-2500 E (Kinematica, Lucerne, Switzerland), and the pH value was determined using a pH meter (Orion Star A211, Thermo Fisher Scientific, Inc., Waltham, MA, USA). The pH of the sample homogenate was measured at a temperature of 18–25 °C.

### 2.4. Instrumental Color Analysis

The instrumental color analysis was conducted as described by the International Commission on Illumination (CIE) [21]. Before color measurement, a cooked sample for each treatment was placed in the flat surface and left at room temperature for 10 min. Meat color was assessed using a colorimeter (CR-300; Minolta Co., Osaka, Japan) to determine the lightness (L*), redness (a*), and yellowness (b*) of the cut surface of the sample. Prior to the measurement, the instrument was calibrated using a standard white plate (Y = 93.60, x = 0.3134, y = 0.3194). For every sample, five measurements were obtained.

### 2.5. Instrumental Texture Analysis

The shear force values were measured using a Texture Analyzer TA 1 (NEXYGENPlus™ software version 3.0, Lloyd Instruments, Fareham, UK) equipped with a 500 N load cell and operated at a test speed of 50 mm/min. Samples were left at room temperature (20–25 °C) for 10 min, until a constant temperature was reached, before the measurement. The samples prepared for measurement were cut to a length of 3 cm in the fiber direction, a width of 1 cm, and a height of 2 cm. The V blade was placed perpendicular to the fiber direction, and the measurements were conducted.

A texture profile analysis (TPA) was conducted, with slight modification, as previously described [22,23]. The TPA was instrumentally performed to determine the springiness, hardness, chewiness, gumminess, and cohesiveness of the samples (LLOYD instruments, Fareham, UK). Samples cooked under each condition were cut into 1.5 cm × 1.5 cm × 2.5 cm squares for testing. Using a two-cycle compression test with a cross speed of 100 mm/min and a wait time of three seconds, samples were axially compressed to 75% of their initial height. The cylindrical probe used was a of 30 mm in diameter and 12 mm in height.

### 2.6. Measurement of Myofibril Fragmentation Index (MFI)

MFI measurements were conducted as described by Culler et al. [24]. Samples (2 g) cooked under each condition were homogenized twice for 30 s, at 15,000 rpm, in 40 mL of MFI buffer (20 mM phosphate buffer, 1 mM EDTA, 100 mM KCl, pH 7.0). After homogenization, the suspension was centrifuged at 1000× *g* at 4 °C for 15 min. The protein concentration of the suspension was adjusted to 0.5 mg/mL, using MFI buffer, and the absorbance was measured at a wavelength of 540 nm, using a spectrophotometer. MFI was calculated as 200 times the absorbance of the solution.

### 2.7. Scanning Electron Microscopy (SEM)

The ultramicrostructure was examined using a field-emission scanning electron microscope (FE-SEM-II; JSM-7900F, JEOL Ltd., Seoul, Republic of Korea). Samples prepared under each condition were sliced and placed in 2.5% glutaraldehyde with 0.1 M phosphate-buffered saline (PBS; pH 7.4) for 36 h. Subsequently, the samples were rinsed three times, using 0.1 M PBS (pH 7.4), and dehydrated using a gradient of ethanol solutions (30, 50, 70, 90, and 100%) before being frozen at −20 °C. After that, the samples were lyophilized and sputter-coated with platinum, using a Cressington sputter coater 208HR (Cressington Scientific Instruments Ltd., Watford, UK) at 20 MA. The FE-SEM controlled by PC-SEM software version 4.0, with an ×1000 magnification was used to view the microstructure.

### 2.8. Sodium Dodecyl Sulfate–Polyacrylamide Gel Electrophoresis (SDS-PAGE)

Myofibrillar proteins were extracted as previously described [25]. Buffer A was prepared and mixed with 5 g of the sample, which was then homogenized at 12,000 rpm (2 × 30 s) and centrifuged at 2000× *g* (4 °C, 10 min) to remove the supernatant. The pellet was resuspended in 25 mL of buffer A and centrifuged twice. The supernatant was carefully discarded, and the pellet was resuspended in 25 mL of the prepared buffer B, followed by three additional centrifugations. Finally, the suspension buffer was added. The extracted myofibrillar protein was quantified using a bicinchoninic acid protein assay kit (Sigma Chemical Co., St. Louis, MO, USA). An equivalent volume of 20 μg of myofibrillar protein concentration combined with 7 μL of Laemmli’s SDS-sample Buffer (GenDEPOT Co., Katy, TX, USA), which is 4 times concentratedwere heated to 95 °C for ten minutes. Next, each lane of the precast gel (GenScript, 10% polyacrylamide) was loaded with 6 µL of molecular standard marker (Xpert 2 Prestained Protein Marker GenDEPOT Co., Katy, TX, USA). Then, protein was separated by electrophoresis, using a 10% running gel and a 6% stacking gel at 300 V and 30 mA for 90 min (myPowerII300 AE8135, Atto Corp., Tokyo, Japan). Following the separation process, the gel was stained overnight with Coomassie Brilliant Blue (0.1%) and repeatedly decolored with distilled water three times, until the bands became visible.

### 2.9. Microbiological Analysis

Petrifilm (Aerobic Count Plates, Coliform/*E. coli* Count Plates, 3M Company, St. Paul, MN, USA) was used to determine the total aerobic bacterial and *E. coli*/coliform counts. Three grams of each sample was collected and placed in a sterile bag, along with 27 mL of sterile saline solution. A stomacher (BagMixer 400; Interscience, Saint-Nom la Bretèche, France) was used to homogenize the samples for 1 min. One milliliter of the homogenate was inoculated into a Petri film after dilution with sterile saline solution. Petrifilm was cultured at 37 °C for 48 h, and the colony count was determined. The result was presented in Log CFU/g.

### 2.10. Measurement of VOC

A five-time repetition of sample for each treatment was analyzed for VOC. Headspace solid-phase microextraction (HS-SPME) was employed to extract the VOC following the method described by Garcia-Esteban et al. [26]. Samples weighing 5 g each were placed in 20 mL SPME vials and sealed with septa and crimp caps. The sealed vials were equilibrated for 25 min in a constant-temperature water bath at 60 °C to establish an equilibrium state for the VOCs in the headspace. An SPME fiber (50/30 µm DVB/Carboxen/PDMS, Supelco, Bellefonte, PA, USA) was exposed to the headspace and allowed to adsorb VOCs for 30 min. The VOCs adsorbed onto the SPME fibers were analyzed using an Agilent 8890 GC/5977B MSD (Agilent Technologies, Palo Alto, CA, USA). Agilent GC/MS Enhanced MassHunter software (version 10.0.368) was used for device operation, and Aligent MassHunter software was used to collect and analyze data. Helium was used as the carrier gas, and a DB-5MS column (30 m × 0.25 mm id, 0.25 μm) was used as the analytical column, at a flow rate of 1.3 mL/min. The injection port temperature was set at 250 °C, and the detector temperature at 280 °C. The oven temperature was maintained at 40 °C for 10 min in gradient mode, followed by an increase to 250 °C at a rate of 5 °C/min and a 5-minute hold. The aroma compounds adsorbed onto the SPME fibers were desorbed by exposure to the injector for 15 min. The aroma compounds present in the samples were identified based on the linear retention index, using alkane standards (C10–C26) and mass spectral library data [27]. The concentration of the aroma compounds was expressed in area units (a.u.) × 106. Additionally, the flavor characteristics of each aroma compound were determined using FlavorDB (https://cosylab.iiitd.edu.cn/flavordb/ (accessed on 5 October 2023) and FooDB Flavornet, as described by Garg et al. [28].

### 2.11. Analysis of Nucleotide-Related Compounds

An analysis of the nucleotide-related compounds was conducted by using high-performance liquid chromatography (HPLC) based on the method of Jayasena et al. [29]. Five grams of the sample was homogenized after adding 25 mL of 0.7 M perchloric acid (PCA) solution. The homogenate was then centrifuged at 2000× *g* for 15 min, at 0 °C. The supernatant was collected via filtration through Whatman filter paper No. 4, and the remaining precipitate was subjected to the same procedure, with an additional 25 mL of 0.7 M PCA solution. The collected supernatant was then mixed. The pH of the filtered supernatant was adjusted to 6.5, using a 5 N KOH solution. Once the adjustment was complete, the samples were left at 0 °C for 30 min. Subsequently, it was thoroughly mixed, and the upper layer was then filtered using a 0.22 μm syringe filter for subsequent HPLC analysis. An Agilent Infinity 1260 instrument (Agilent Technologies, Santa Clara, CA, USA) was used for analysis. The column used was a Nova-Pak C18 column (3.9 × 150 mm, 4 μm; Waters, Milford, USA). The column temperature was maintained at 40 °C. The mobile phase consisted of 1% trimethylamine in phosphoric acid (pH 6.5). The flow rate was set at 1 mL/min, and the injection volume was 10 μL. The detector was operated at a wavelength of 254 nm. Nucleotide-related compounds in the samples were identified via a comparison with the retention time of ATP, ADP, AMP, IMP, GMP, inosine, and hypoxanthine standard (Sigma Aldrich Co., St. Louis, MO, USA). The concentrations were quantified based on the calibration curve, and the data were presented as mg/100 g of fresh sample. The Agilent OpenLAB CDS Chemstation Edition 2.16.12 software was used to operate the equipment.

### 2.12. Analysis of Fatty Acids

For the fatty acid analysis, the fats were extracted using a Folch solution (chloroform: methanol; 2:1). The sample was mixed with 1.5 mL of 0.5 N NaOH and methanol. The mixture was vortexed and heated for 5 min at 100 °C, cooled in cold water, and mixed with 2 mL of BF_3_-methanol solution (approximately 10%, Supelco, Bellefonte, PA, USA). The mixture was vortexed and heated at 100 °C for 2 min prior to cooling. To extract fatty acid methyl esters, 2 mL of isooctane was added, and the mixture was vortexed for 1 min. The mixture was then mixed with 1 mL of saturated NaCl solution and vortexed thoroughly for 1 min. The layers were separated using a centrifuge (1000× *g*, 3 min), and the upper layer containing the fatty acid methyl esters (iso-octane layer) was transferred to a gas chromatography vial for analysis. The analysis was conducted using gas chromatography Agilent (FID-7890B) equipment (Agilent Technologies, Santa Clara, CA, USA). The column used for the analysis was an Omegawax 250 (30 m × 0.25 mm id, 0.25 μm film thickness; Supelco, Bellefonte, PA, USA). The Agilent OpenLAB CDS Chemstation for GC Edition B.01.01.069 software was used to operate the equipment. A flame ionization detector was used, helium (99.99% purity) was used as a carrier gas, the column flow rate was set at 1.2 mL/min, and the split ratio was 100:1, with an injection volume of 1 μL. The injection port temperature was set to 250 °C, while the detector temperature was set to 260 °C. The oven temperature was initially set at 150 °C for 2 min and then raised at a rate of 4 °C/min to reach 220 °C for 30 min. By comparing the retention durations and peak areas of the fatty acid external standard (ESTD) PUFA No. 2 (Animal Source) 47015-U from Supelco (Bellefonte, USA), the fatty acid peaks were found. The result was presented as the fatty acid percentage composition of total fat.

### 2.13. Statistical Analysis

The data were examined using a one-way ANOVA, and a general linear model was applied to evaluate the effect of the cooking temperature and time, and their interaction as the fixed effect on different parameters. Tukey’s analysis was conducted to test differences between means at *p* < 0.05. All statistical values were presented as mean and standard error of the mean (SEM). The analyses were performed using the Statistical Analysis System software ver. 9.4 (SAS Institute Inc., Cary, NC, USA). MetaboAnalyst 5.0, an online software package, was used to perform heatmap and partial least squares discriminant analyses (PLS-DAs) [30].

## 3. Results and Discussion

### 3.1. Proximate Components and Cooking Losses

The proximate components and cooking loss of horse loin cooked using the sous-vide method at different cooking durations and temperatures are summarized in Table 1. The proximate composition of conventionally boiled meat (control group) and sous vide-cooked meat included moisture content ranging from 62.64 to 67.25%, crude protein from 26.38 to 30.72%, crude fat from 6.14 to 8.49%, ash content from 1.04 to 1.23%, and cooking loss from 22.88 to 37.58%. No significant differences were observed between the control and treatment groups in terms of moisture, crude fat, or ash content. The treatment groups had significantly higher crude protein content than that in the control group (26.38%), except for the 65 °C for 12 h treatment group (28.14%), and the protein composition of the treatment groups was influenced by the cooking temperature (*p =* 0.0099). The study revealed that the crude protein content was 19.8–21.17%, slightly higher than the values reported by Lorenzo et al. [31], but similar to the 28.1% collected by the United States Department of Agriculture [32]. The cooking loss in all treatment groups was significantly higher than that in the control group (22.88%), and the cooking loss in the treatment groups was affected by the sous-vide cooking temperature (*p =* 0.0003). As the temperature increased, there was a corresponding increase in weight loss. The results of this study were generally consistent with those previously reported in studies on beef [8,33], lamb [8], and pork [34]. Myofibrillar proteins retain most of the moisture, and according to Roldán et al. [8], these proteins denature and shrink as the temperature rises between 40 °C and 90 °C. Additionally, collagen will shrink at temperatures ranging from 56 to 62 °C, leading to an increase in cooking loss.

### 3.2. Changes in pH and Instrumental Color

Table 2 presents the pH values and color coordinates for the sous vide-cooked samples. The pH values of the control group and sous vide-cooked meat ranged from 6.20 to 6.40. The L* values ranged from 46.41 to 48.70, a* values from 13.60 to 16.95, and b* values from 12.96 to 14.31. The pH values in all treatment groups, except for the 65 °C for 12 h treatment group (6.34), were significantly higher than those in the control group (6.20). Meat pH generally increases after cooking, owing to the fragmentation of bonds involving sulfhydryl, imidazole, and hydroxyl groups; the alterations in the electric charge of the acidic groups; the separation of the peptide chain; and the production of new alkaline compounds during cooking [15].

There were no significant differences in L* values between the control and treatment groups. However, the L* values among the treatment groups were influenced by the cooking temperature (*p =* 0.0037) and cooking time (*p =* 0.0479). A significant difference in the L* value within the treatment group was found, with the L* value of the 65 °C for the 24 h treatment group being higher than that of the 70 °C for 18 h treatment group. The high L* value of meat cooked at a low temperature may be due to increased free water remaining on the surface of the sliced sample prior to the color analysis [34]. Samples cooked at low temperatures, thereby preserving increased moisture, exhibited water release on the surface during slicing for color analysis, whereas samples that lost a great amount of moisture during cooking appeared to have low exuded water levels on the surface [8].

The a* values in the 70 °C for 18 h (13.84) and the 70 °C for 24 h (13.60) treatment groups were significantly lower than those in the control (16.95) and the 65 °C for 12 h (16.29) treatment groups. The a* values among the treatment groups were affected by the cooking temperature (*p <* 0.0001) and cooking time (*p =* 0.0194). These findings suggest that myoglobin degradation increases with the cooking temperature. Studies have reported a decrease in redness as the cooking temperature increases for beef [35], pork [34], and lamb [8]. The extent of myoglobin denaturation was found to be associated with the cooking temperature, which resulted in an increased degree of myoglobin denaturation, consequently leading to decreased a* values [36,37].

The b* values in the 65 °C for 24 h (13.91) and the 70 °C for 24 h (14.31) treatment groups were significantly higher than those in the control group (12.96), and the b* values among the treatment groups were influenced by the sous-vide cooking duration (*p =* 0.0004). A trend of decreasing a* values and increasing b* values with a long sous-vide cooking duration was observed by Christensen et al. [38], García-Segovia et al. [35], and Roldán et al. [8]. This can be attributed to the formation of metmyoglobin and the subsequent heat-induced denaturation of this protein, resulting in the development of a brownish hue.

### 3.3. Changes in Instrumental Texture

The values of textural variables, including shear force, TPA, and MFI, for horse loins cooked using the sous-vide method at different cooking times and temperatures are presented in Table 3. The shear force represents the force required to cut meat against the muscle fiber direction. The shear force values at 65 °C for 24 h (36.36 N) and 70 °C for 24 h (35.70 N) were significantly lower than those in the control group (52.64 N), indicating an improvement in tenderness due to cooking. Furthermore, the shear force among the treatment groups was influenced by the sous-vide cooking duration (*p =* 0.0012).

TPA includes hardness, which represents the force required for deformation; and chewiness, which represents the force needed for chewing until a substance is swallowed. No significant differences were observed in hardness and chewiness between the control and treatment groups. However, the cohesiveness, springiness, and gumminess values of the treatment group, particularly those cooked at 70 °C for 18 and 24 h, were significantly different from those of the control group. Springiness was influenced by the sous-vide cooking duration (*p =* 0.0074) and the interaction between temperature and time (*p =* 0.0003).

The instrumental texture analysis revealed that the cooking time and the interaction between cooking time and temperature had varying degrees of influence on the textural characteristics of horse loins cooked using the sous-vide method. The results of instrumental texture analyses, including shear force and TPA, indicated that the sous-vide cooking method improved the tenderness of horse loins. These findings are consistent with those reported by Roldán et al. [8], who reported that the sous-vide method increased the tenderness of lamb loin with a prolonged cooking time.

Meat tenderness during cooking is associated with heat exposure. Heat solubilizes connective tissue, thereby increasing meat tenderness; however, the denaturation of myofibrillar proteins may result in tough meat [39]. Additionally, water loss from the muscle tissue upon heating contributes to meat toughening. The transition from a viscoelastic material to an elastic material also influences texture changes during heating [12]. Raw meat is difficult to chew because of the high viscosity flow between fibers, but heating to 65 °C improves tenderness, as sarcoplasmic proteins bind together to form a gel, making it accessible for chewing [12]. However, temperatures exceeding 65 °C up to 80 °C result in tough meat due to an increased elastic modulus [40]. After 6 h, the residual collagenolytic activity at 60 °C may explain the tenderness of meat cooked at these temperatures for extended periods [12].

Our study observed a tendency for the hardness and shear force to decrease with a prolonged cooking duration. This phenomenon could be attributed to the increased solubilization of collagen during extended cooking duration, whereas myofibrillar shrinkage may have already reached its maximum and showed no further significant increase over increased durations. The enhanced tenderness observed during the extended cooking of horse loin in our study, as indicated by the shear force and TPA values, may be attributed to the extensive breakdown of the perimysium surrounding the muscle bundles [12].

MFI functions, as an indicator of muscle fiber integrity, and an increase in MFI are associated with the breakdown of muscle fiber proteins. Meat tenderness is influenced not only by the distribution and quantity of connective tissues but also by an increased level of meat proteolysis, as indicated by the MFI [41,42]. The findings of the present study align with these arguments. A significant decrease in the shear force and a significant increase in the MFI were observed as the sous-vide temperature and duration increased. The MFI in the control group was 41.15, whereas, in the 70 °C treatment groups, it reached 55.03 at 12 h, 56.71 at 18 h, and 52.49 at 24 h, showing significantly high values. The MFI among the treatment groups was significantly influenced by the sous-vide cooking temperature (*p =* 0.0145). This result was consistent with the findings of Karki et al. [42], who also reported that the sous-vide temperature and duration affected the MFI.

### 3.4. Changes in Microstructure

The microstructural changes in sous vide-cooked horse loin at different cooking times and temperatures are shown in Figure 1. In raw samples, the connective tissue surrounding the muscle fibers remained unaltered, making it difficult to distinguish individual muscle fibers. However, in the sous vide-treated samples at 65 °C and in the control group, gaps between the muscle fibers became visible. At a temperature of 70 °C, the horse meat exhibited an increased density and a tightly compacted fiber arrangement in its structure. This phenomenon is consistent with the findings of Roldán et al. [8], who observed similar gaps between muscle fibers following the sous-vide cooking of lamb meat.

Figure 1 also reveals the presence of granular deposits within the spaces between the muscle fibers. At 70 °C, these granular deposits not only occupy the spaces between muscle fibers but also extend into the endomysium and between fiber bundles. The connective tissue undergoes denaturation, resulting in gel formation. Continued heating leads to the creation of a denatured collagen gel surrounding each muscle fiber, which may contribute to maintaining cell shape despite water loss [8]. A similar observation was reported by Li et al. [43] in their research on beef muscle cooked in the temperature range of 60–80 °C. They also reported significant granulation at 70 °C, possibly attributed to sarcolemma distortion.

### 3.5. Changes in Myofibrillar Protein

Figure 2 presents the results of the electrophoresis of myofibrillar proteins in sous vide-cooked horse loin at different cooking durations and temperatures. The quality characteristics of muscle foods are significantly influenced by the functionality of myofibrillar proteins. The cooking process can alter the functional attributes of these proteins, leading to changes in texture and microstructure. The intensity of the band represents the degree of protein degradation. Thinner bands mean that the protein is more degraded compared to the thicker bands. Myofibrillar proteins, as described by Yin et al. [9], constitute 55–60% of the total muscle proteins and play a significant role in enhancing meat tenderness during denaturation. A comparison of the myosin heavy chain (MHC) bands between the control group and the treated groups reveals that the MHC in the 70 °C for 18 h treatment group has undergone significant degradation, resulting in the thinnest band. The intensity of the tropomyosin β-chain and troponin-T decreased after sous-vide cooking at 70 °C for 18 h, compared to the control or raw meat. This suggests that sous-vide cooking breaks down the peptides into smaller ones. The functionality of myofibrillar proteins significantly influences meat quality attributes, including texture and microstructure, which can be altered during cooking [44].

### 3.6. Changes in Microbiology

The change in microbiology, including total aerobic bacteria, coliform, and *E. coli*, was summarized in Table 4. In all treatment groups, the total aerobic bacterial count was not detected, except in the 65 °C for 24 h treatment group. However, this count was significantly lower than that in the control group. Coliforms and *E. coli* were not detected in any treatment groups. These findings align with those of previous studies that have reported minimal microbiological counts or the absence of such counts in sous video-cooked pork [45] and lamb samples [8]. A review by Baldwin [12] provided pasteurization timings for sous video-cooked meat in accordance with the Food and Drug Administration regulations. The review affirms that even the shortest time–temperature combination used in our study (65 °C for 12 h) was more than sufficient for pasteurizing meat.

### 3.7. Changes in Volatile Compounds

Volatile organic compounds can be sorted into various categories including acids, alcohols, and aldehydes. Each of these groups contributes to the distinct flavors found in meat [46]. A total of 83 volatile compounds were detected in the sous vide-cooked horse loin and are summarized based on cooking duration and temperature in Table 5. These volatile compounds were categorized into the following chemical families: 7 alcohols, 24 aldehydes, 15 esters, 25 hydrocarbons, and 12 ketones. The total number of compounds detected differed slightly from those found in previous horse meat (foal) samples [47,48]. Hydrocarbon compounds were the most abundant, with 1-octen-3-ol being the most prominent alcohol known for its distinctive aroma, which is reminiscent of fishiness, fattiness, mushrooms, and grassiness [49]. Decanal dominated the aldehyde category, imparting soap and orange-peel aromas [50], as well as a tallow-like scent. Among the esters, n-caproic acid vinyl ester was the most prevalent, whereas cyclotetrasiloxane and octamethyl were the leading compounds in the hydrocarbon category. The most abundant ketone was 3,3-Dimethyl-1,2-epoxybutane. These volatile compounds originate from the breakdown of lipids (alcohols, aldehydes, hydrocarbons, and ketones) or the Maillard reaction [48]. Trends in VOCs resulting from lipid degradation exhibited similar patterns under varying temperatures and cooking conditions. For alcohols, aldehydes, and ketones, the highest values were observed in samples cooked at 65 °C for 18 h, and the lowest levels were observed in those cooked at 70 °C for 24 h. In contrast, for hydrocarbons, the highest level was found at 65 °C for 12 h, and the lowest at 70 °C for 12 h. The raw meat was also analyzed for volatile compounds to distinguish the aroma changes between before and after heat treatment. The process of heating results in the production of new molecules, mostly from lipid oxidation and Maillard browning processes, which are collectively responsible for the desired flavor of the meat. The heating and long cooking processes promote the breakdown of polyunsaturated fatty acids, including linoleic acid, resulting in the development of additional VOCs (octanal, heptanal, 2-octanone, and nonanal) during the late stages of lipid oxidation; some of these carbonyl compounds may interact with amino acids such as lysine, cysteine, and glutathione [7].

The results of heatmap and PLS-DA are presented in Figure 3. Through the heatmap, it can be observed that the 65 °C for 12 h and 65 °C for 18 h treatment groups share a similar composition, distinguishing them from the rest of the treatment groups. PLS-DA identified 25 volatile compounds with variable importance in projection (VIP) scores above 1.2, with 2-undecenal having the highest VIP score and contributing to fresh, fruity, and orange-peel aromas [51].

### 3.8. Change in Nucleotide-Related Compound

Adenosine triphosphate (ATP), adenosine diphosphate, adenosine monophosphate (AMP), inosine monophosphate (IMP), inosine, guanosine monophosphate (GMP), and hypoxanthine in the sous video-cooked horse loin are presented in Table 6. Nucleotide levels were significantly influenced by the cooking temperature, particularly the ATP, AMP, and IMP levels (*p <* 0.001). Both the temperature and cooking time had significant effects on the ATP levels (*p <* 0.001). These findings are consistent with those of previous research, which also indicated that temperature, rather than cooking time, significantly affects nucleotide content [52]. IMP and GMP are nucleotides that contribute to the umami taste of meat [53]. The IMP content in the 65 °C for 12 h (22.58), 65 °C for 18 h (25.00), and 65 °C for 24 h (21.35) treatment groups was significantly lower than that in the control group (42.35). However, the 70 °C for 18 h treatment group showed a higher IMP content than that in all of the 65 °C treatment groups. Temperature has a significant impact on IMP degradation, which can be further explained by two potential mechanisms: first, IMP dephosphorylation into inosine; and second, the involvement of IMP and ribose as flavor precursors in various secondary reactions, resulting in the production of heterocyclic volatile compounds [52].

### 3.9. Changes in Fatty Acids

The changes in fatty acid composition of the sous vide-cooked horse loin based on cooking duration and temperature are presented in Table 7. This study found no significant differences in the proportions of all analyzed fatty acids between the control and treatment groups. This suggests that the sous-vide cooking process, regardless of the specific parameters used, does not significantly alter the fatty acid composition of horse loin. Interestingly, our study revealed that linoleic acid was the most abundant fatty acid in the sous vide-cooked horse loin. This contrasts with the findings of Seong et al. [54], who reported that, in the loin cut of native Jeju horses, oleic acid was the most abundant, accounting for 32.78% of total fatty acids, followed by palmitic acid (31.16%) and then linoleic acid (16.42%). This discrepancy could be attributed to differences in the breed of horse, diet, or other factors that can influence the fatty acid composition of meat. Furthermore, our study found that the ratios of monounsaturated fatty acids to saturated fatty acids (MUFAs/SFAs) and polyunsaturated fatty acids to saturated fatty acids (PUFAs/SFAs) did not change significantly after sous-vide cooking. This was consistent across all treatment groups, indicating that these ratios are not affected by the sous-vide cooking process. According to the diversity in the fatty acid content of triglyceride molecules among different livestock species, pig fat and horse fat are the most sensitive to oxidative breakdown, with sheep tallow being somewhat less susceptible and cow tallow displaying the highest resilience [55]. Horse muscle contains a high concentration of unsaturated fatty acids (60.49–63.04%), with monounsaturated acids (38–55%) and palmitoleic (3–10%) being the most prevalent, accounting for up to 45.16% of all fatty acids [56]. This resilience in the ratios indicates that the sous-vide method may preserve the relative proportions of these fatty acids, contributing to the maintenance of the nutritional quality of the cooked horse loin. This is an important finding, as these ratios are often considered in nutritional analyses of meat.

## 4. Conclusions

Sous-vide cooking has been found to enhance the quality of horse loin meat, particularly in terms of tenderness and flavor, when compared to conventionally boiled meat. Treatments at both 65 °C and 70 °C for 24 h resulted in substantial improvements in tenderness, as indicated by reduced shear force values. It is noteworthy that the tenderness of the meat, as measured by shear force, is primarily influenced by the duration of sous-vide cooking, rather than the temperature. In addition to improving tenderness, sous-vide cooking also resulted in a distinctive aroma profile for horse meat. This profile was characterized by unique volatile organic compounds (VOCs) such as 1-octen-3-ol, decanal, n-caproic acid vinyl ester, cyclotetrasiloxane, octamethyl, and 3,3-dimethyl-1,2-epoxybutane. These VOCs can serve as markers to distinguish sous vide-cooked horse meat from meat cooked using other methods. Moreover, sous-vide cooking significantly reduced the total aerobic bacteria in the horse meat, further enhancing its quality. However, further research is needed to ascertain whether these VOCs are unique to sous-vide cooking or if they are also produced by other cooking methods. This study provides a foundation for future investigations into the effects of sous-vide cooking on the quality properties of horse meat.

## Figures and Tables

**Figure 1 foods-13-00280-f001:**
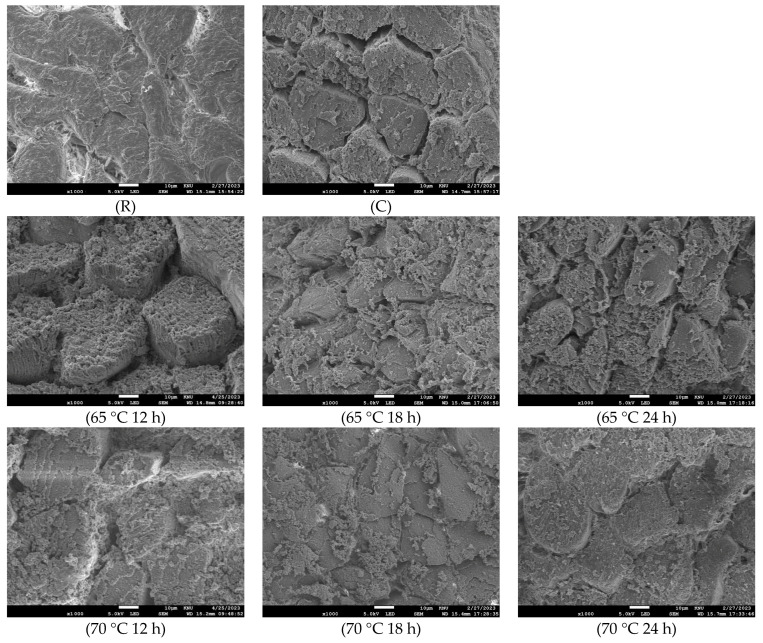
Microstructure of raw and horse loin cooked sous vide at different conditions (1000× magnification). R, raw meat. C, control (boiled at 72 °C for 40 min).

**Figure 2 foods-13-00280-f002:**
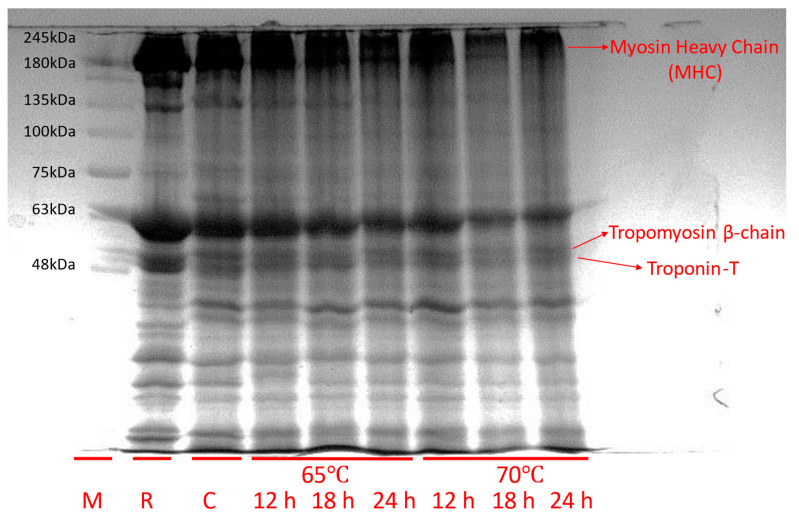
Myofibrillar protein profile (SDS-PAGE) of raw and horse loin cooked sous vide at different conditions. M, size marker. R, raw meat. C, control (boiled at 72 °C for 40 min).

**Figure 3 foods-13-00280-f003:**
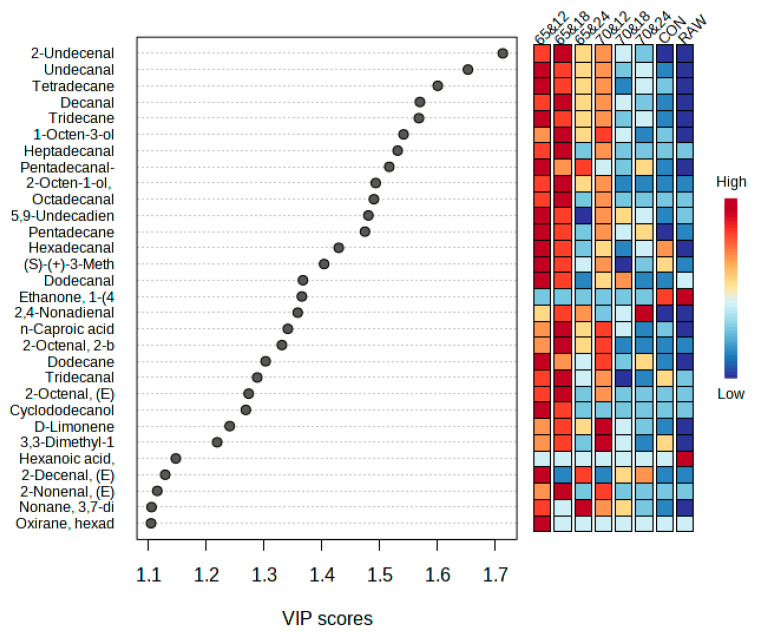
Heatmap analysis and partial least square discriminant analysis (PLS-DA) from volatile organic compounds of raw and horse loin cooked sous vide at different conditions—the important flavors (VIP ≥ 1.2) screened by PLS-DA. Raw: raw meat. Con: control (boiled at 75 °C for 40 min).

**Table 1 foods-13-00280-t001:** Proximate composition (%) and cooking loss (%) of horse loin cooked sous vide at different conditions.

Parameter	C	65 °C			70 °C			SEM	P (T)	P (t)	P (T*t)
		12 h	18 h	24 h	12 h	18 h	24 h				
Moisture ^NS^	67.25	65.21	64.46	64.31	62.64	62.68	63.54	1.125	0.0834	0.9384	0.7428
Crude protein	26.38 ^c^	28.14 ^bc^	29.57 ^ab^	29.46 ^ab^	29.92 ^ab^	30.72 ^a^	30.41 ^ab^	0.559	0.0099	0.1379	0.7518
Crude lipid ^NS^	6.14	8.31	7.30	7.45	7.39	8.49	7.59	0.535	0.7738	0.7709	0.2016
Crude ash ^NS^	1.13	1.21	1.21	1.13	1.14	1.23	1.04	0.086	0.5182	0.3432	0.8211
Cooking loss	22.88 ^c^	32.09 ^b^	32.97 ^ab^	33.68 ^ab^	37.28 ^ab^	37.58 ^a^	35.95 ^a^	1.229	0.0003	0.8768	0.4477

^NS^ Non-significant. C, control (boiled at 72 °C for 40 min). ^a–c^ Means within the same row with different superscript letters differ significantly (*p* < 0.05). SEM, standard error of means. T, temperature. t, time. T*t, The interaction between cooking time and temperature.

**Table 2 foods-13-00280-t002:** pH and instrumental color of horse loin cooked sous vide at different conditions.

Parameter		C	65 °C			70 °C			SEM	P (T)	P (t)	P (T*t)
			12 h	18 h	24 h	12 h	18 h	24 h				
pH		6.20 ^b^	6.34 ^ab^	6.36 ^a^	6.36 ^a^	6.40 ^a^	6.34 ^a^	6.37 ^a^	0.032	0.5301	0.7991	0.4176
Color	L*	46.99 ^ab^	48.23 ^ab^	48.56 ^ab^	48.70 ^a^	46.70 ^ab^	46.41 ^b^	48.51 ^ab^	0.515	0.0037	0.0479	0.1518
	a*	16.95 ^a^	16.29 ^a^	15.57 ^ab^	15.03 ^ab^	14.58 ^ab^	13.84 ^b^	13.60 ^b^	0.553	<0.0001	0.0194	0.9028
	b*	12.96 ^c^	13.30 ^bc^	13.40 ^bc^	13.91 ^ab^	13.50 ^bc^	13.61 ^abc^	14.31 ^a^	0.158	0.0586	0.0004	0.8159

C, control (boiled at 72 °C for 40 min). ^a–c^ Means within the same row with different superscript letters differ significantly (*p* < 0.05). SEM, standard error of means. T, temperature. t, time. T*t, The interaction between cooking time and temperature.

**Table 3 foods-13-00280-t003:** Instrumental texture parameters (shear force values and TPA) and MFI of horse loin cooked sous vide at different conditions.

Parameter	C	65 °C			70 °C			SEM	P (T)	P (t)	P (T*t)
		12 h	18 h	24 h	12 h	18 h	24 h				
Shear force (N)	52.64 ^a^	50.72 ^ab^	44.65 ^ab^	36.36 ^b^	50.57 ^ab^	44.45 ^ab^	35.70 ^b^	3.426	0.9080	0.0012	0.9969
Hardness (N)	165.78	142.72	151.83	130.29	158.79	131.98	140.66	10.984	0.8164	0.4217	0.2607
Cohesiveness	0.45 ^a^	0.37 ^bc^	0.41 ^ab^	0.38 ^bc^	0.41 ^abc^	0.38 ^bc^	0.36 ^c^	0.012	0.6640	0.0889	0.0108
Springiness (cm)	0.76 ^b^	0.89 ^ab^	0.73 ^b^	0.95 ^a^	0.87 ^ab^	0.94 ^a^	0.93 ^a^	0.036	0.0217	0.0074	0.0003
Gumminess (N)	76.64 ^a^	54.24 ^abc^	67.77 ^abc^	49.93 ^abc^	74.44 ^ab^	47.91 ^bc^	46.05 ^c^	6.417	0.8334	0.0690	0.0212
Chewiness (N) ^NS^	56.76	52.78	48.81	46.46	66.28	44.65	40.87	6.672	0.8281	0.0711	0.3289
MFI	41.15 ^b^	50.75 ^ab^	50.78 ^ab^	47.59 ^ab^	55.03 ^a^	56.71 ^a^	52.49 ^a^	2.413	0.0145	0.2789	0.9405

^NS^ Non-significant. C, control (boiled at 72 °C for 40 min). ^a–c^ Means within the same row with different superscript letters differ significantly (*p <* 0.05). SEM, standard error of means. T, temperature. t, time. T*t, The interaction between cooking time and temperature.

**Table 4 foods-13-00280-t004:** The change in microbiology (Log CFU/g) for raw and horse loin cooked sous vide at different conditions.

Parameter	R	C	65 °C			70 °C			SEM
			12 H	18 H	24 H	12 H	18 H	24 H	
Total aerobic bacteria	2.84 ^a^	0.00 ^b^	0.00 ^b^	0.00 ^b^	0.17 ^b^	0.00 ^b^	0.00 ^b^	0.00 ^b^	0.608
Coliforms	N.D.	N.D.	N.D.	N.D.	N.D.	N.D.	N.D.	N.D.	-
*E. coli*	N.D.	N.D.	N.D.	N.D.	N.D.	N.D.	N.D.	N.D.	-

R, raw meat; C, control (boiled at 72 °C for 40 min). ^a,b^ Means within same row with different superscript letters differ significantly (*p* < 0.05). SEM, standard error of means. N.D., not detected.

**Table 5 foods-13-00280-t005:** Selected volatile compounds (in area units × 10^6^) of raw and horse *M. longissimus thoracis et lumborum* cooked sous vide at different conditions.

Compound	R	C	65 °C			70 °C			SEM
			12 h	18 h	24 h	12 h	18 h	24 h	
(S)-(+)-3-Methyl1-pentanol	0.06 ^b^	0.18 ^b^	0.63 ^a^	0.53 ^a^	0.10 ^b^	0.45 ^a^	0.00 ^b^	0.07 ^b^	0.052
1-Octen-3-ol	0.66 ^d^	2.91 ^cd^	16.66 ^b^	26.87 ^a^	4.84 ^c^	16.97 ^b^	3.40 ^cd^	1.62 ^cd^	0.913
2-Octen-1-ol, (E)-	0.00 ^c^	0.00 ^c^	0.88 ^b^	1.30 ^a^	0.15 ^c^	0.60 ^b^	0.00 ^c^	0.00 ^c^	0.091
Ethanone, 1-(4,5-dihydro-2-thiazolyl)-	0.15 ^a^	0.08 ^b^	0.00 ^c^	0.00 ^c^	0.00 ^c^	0.00 ^c^	0.00 ^c^	0.00 ^c^	0.013
2-Decenal, (E)-	0.00 ^d^	0.00 ^d^	0.05 ^a^	0.00 ^d^	0.02 ^b^	0.00 ^d^	0.01 ^cd^	0.02 ^bc^	0.003
2-Octenal, (E)-	0.00 ^c^	0.00 ^c^	0.02 ^b^	0.03 ^a^	0.00 ^c^	0.01 ^bc^	0.00 ^c^	0.00 ^c^	0.003
2-Octenal, 2-butyl-	0.00 ^c^	0.00 ^c^	0.06 ^b^	0.18 ^a^	0.02 ^c^	0.07 ^b^	0.00 ^c^	0.00 ^c^	0.006
2-Undecenal	0.00 ^d^	0.00 ^d^	3.82 ^a^	4.42 ^a^	1.29 ^b^	3.69 ^a^	1.11 ^bc^	0.55 ^cd^	0.160
Cyclododecanol	0.00 ^b^	0.00 ^b^	0.08 ^a^	0.01 ^b^	0.00 ^b^	0.00 ^b^	0.00 ^b^	0.00 ^b^	0.003
Decanal	0.01 ^e^	0.02 ^e^	3.97 ^b^	8.05 ^a^	1.76 ^c^	3.81 ^b^	1.21 ^cd^	0.57 ^de^	0.257
Dodecanal	0.00 ^c^	0.00 ^c^	0.10 ^a^	0.03 ^b^	0.00 ^c^	0.01 ^bc^	0.01 ^bc^	0.00 ^c^	0.004
Hexadecanal	0.02 ^c^	0.14 ^c^	0.88 ^a^	0.36 ^b^	0.08 ^c^	0.14 ^c^	0.07 ^c^	0.08 ^c^	0.042
Octadecanal	0.00 ^d^	0.00 ^d^	0.08 ^b^	0.12 ^a^	0.00 ^d^	0.03 ^c^	0.00 ^d^	0.00 ^d^	0.005
Pentadecanal-	0.02 ^c^	0.05 ^c^	4.06 ^a^	1.26 ^b^	1.31 ^b^	0.86 ^bc^	0.44 ^bc^	1.12 ^bc^	0.240
Tridecanal	0.01 ^de^	0.02 ^cd^	0.05 ^b^	0.07 ^a^	0.01 ^cde^	0.03 ^c^	0.00 ^e^	0.01 ^e^	0.004
Undecanal	0.00 ^c^	0.01 ^c^	0.62 ^a^	0.41 ^b^	0.07 ^c^	0.11 ^c^	0.05 ^c^	0.05 ^c^	0.028
2,4-Nonadienal, (E,E)-	0.00 ^d^	0.00 ^d^	0.30 ^b^	0.36 ^ab^	0.34 ^ab^	0.18 ^c^	0.19 ^c^	0.39 ^a^	0.017
Heptadecanal	0.00 ^c^	0.00 ^c^	0.03 ^a^	0.03 ^a^	0.00 ^c^	0.01 ^b^	0.00 ^c^	0.00 ^c^	0.002
Hexanoic acid, ethyl ester	0.36 ^a^	0.00 ^b^	0.00 ^b^	0.00 ^b^	0.00 ^b^	0.00 ^b^	0.00 ^b^	0.00 ^b^	0.012
n-Caproic acid vinyl ester	0.22 ^e^	1.90 ^de^	12.36 ^c^	36.95 ^a^	6.19 ^d^	21.37 ^b^	4.82 ^de^	1.30 ^e^	1.013
Cyclotetrasiloxane, octamethyl	20.27	25.70	22.38	20.11	21.84	15.11	24.94	24.82	2.860
Dodecane	0.06 ^c^	0.16	0.64 ^a^	0.41 ^ab^	0.25 ^bc^	0.42 ^ab^	0.21 ^bc^	0.33 ^abc^	0.074
Nonane, 3,7-dimethyl	0.00 ^c^	0.20 ^bc^	0.67 ^ab^	0.29 ^abc^	0.84 ^a^	0.46 ^abc^	0.38 ^abc^	0.26 ^bc^	0.121
Oxirane, hexadecyl	0.00 ^b^	0.00 ^b^	0.07 ^a^	0.00 ^b^	0.00 ^b^	0.00 ^b^	0.00 ^b^	0.00 ^b^	0.004
Pentadecane	0.07 ^c^	0.05 ^c^	0.76 ^a^	0.42 ^b^	0.10 ^c^	0.19 ^bc^	0.11 ^c^	0.14 ^c^	0.051
Tetradecane	0.05 ^b^	0.06 ^b^	0.40 ^a^	0.34 ^a^	0.09 ^b^	0.15 ^b^	0.06 ^b^	0.08 ^b^	0.025
Tridecane	0.05 ^b^	0.06 ^b^	0.53 ^a^	0.44 ^a^	0.12 ^b^	0.22 ^b^	0.07 ^b^	0.09 ^b^	0.043
2-Nonenal, (E)-	0.00 ^d^	0.00 ^d^	0.04 ^c^	0.19 ^a^	0.00 ^d^	0.08 ^b^	0.00 ^d^	0.00 ^d^	0.007
3,3-Dimethyl-1, 2-epoxybutane	0.32 ^d^	1.55 ^c^	2.72 ^b^	2.86 ^ab^	1.06 ^cd^	3.89 ^a^	1.21 ^cd^	0.51 ^d^	0.225
5,9-Undecadien-2-one, 6,10-dimethyl-, (E)-	0.01 ^c^	0.01 ^c^	0.12 ^a^	0.08 ^b^	0.00 ^c^	0.02 ^c^	0.02 ^c^	0.01 ^c^	0.007
D-Limonene	0.00 ^d^	0.02	0.11 ^b^	0.11 ^b^	0.09 ^bc^	0.21 ^a^	0.06 ^bcd^	0.03 ^cd^	0.015

R, raw meat. C, control (boiled at 72 °C for 40 min). ^a–e^ Means within the same row with different superscript letters differ significantly (*p <* 0.05). SEM, standard error of means.

**Table 6 foods-13-00280-t006:** Nucleotide-related compound (mg/100 g) of horse loin cooked sous vide at different conditions.

Compound	C	65 °C			70 °C			SEM	P (T)	P (t)	P (T*t)
		12 h	18 h	24 h	12 h	18 h	24 h				
ATP	3.03 ^f^	14.42 ^e^	16.55 ^d^	20.58 ^c^	18.03 ^d^	22.80 ^b^	26.01 ^a^	0.382	<0.0001	<0.0001	0.0208
ADP	1.84 ^a^	0.28 ^b^	0.35 ^b^	0.33 ^b^	0.49 ^b^	0.47 ^b^	0.42 ^b^	0.074	0.0052	0.7630	0.4225
AMP	26.91 ^a^	21.84 ^bc^	21.98 ^bc^	19.83 ^c^	25.50 ^ab^	24.80 ^ab^	24.72 ^ab^	0.823	<0.0001	0.2341	0.4636
IMP	42.35 ^a^	22.58 ^bc^	25.00 ^bc^	21.35 ^c^	29.42 ^abc^	41.02 ^a^	37.27 ^ab^	3.107	<0.0001	0.0761	0.2049
Inosine	45.83 ^a^	44.79 ^a^	38.63 ^ab^	31.86 ^b^	31.13 ^b^	31.69 ^b^	29.61 ^b^	2.195	0.0004	0.0096	0.0378
GMP	0.98 ^ab^	0.79 ^b^	0.68 ^b^	0.73 ^b^	0.89 ^ab^	1.03 ^ab^	1.35 ^a^	0.103	0.0019	0.1706	0.1019
Hypoxanthine	22.16 ^c^	27.99 ^b^	29.30 ^ab^	32.28 ^a^	31.03 ^ab^	31.35 ^a^	32.03 ^a^	0.685	0.0174	0.0092	0.1032

^a–f^ Different superscript letters within the same row mean significantly different between treatments (*p <* 0.05). C, control (boiled at 72 °C for 40 min). ATP, adenosine triphosphate. ADP, adenosine diphosphate AMP, adenosine-5′-monophosphate. IMP, inosine-5′-monophosphate. GMP, guanosine-5′-monophosphate. SEM, standard error of means. T, temperature. t, time. T*t, The interaction between cooking time and temperature.

**Table 7 foods-13-00280-t007:** Fatty acid composition (%) of horse loin cooked sous vide at different conditions (no significant difference in horse loin cooked sous vide under different conditions was found).

Compound ^NS^	C	65 °C			70 °C			SEM	P (T)	P (t)	P (T*t)
		12 h	18 h	24 h	12 h	18 h	24 h				
C14:0 (myristic acid)	2.95	3.65	2.86	3.30	4.65	3.63	3.06	0.493	0.2534	0.1564	0.4709
C16:0 (palmitic acid)	25.10	25.58	25.01	25.25	26.35	25.68	25.47	0.716	0.3351	0.5932	0.9104
C16:1n7 (palmitoleic acid)	2.46	3.12	2.20	2.96	3.26	3.10	2.22	0.464	0.8060	0.4427	0.2930
C18:0 (stearic acid)	8.79	7.01	9.38	7.69	6.67	7.15	8.60	1.064	0.5576	0.3998	0.3980
C18:1n9 (oleic acid)	19.50	21.89	19.76	21.88	19.32	24.23	19.91	2.666	0.9908	0.8771	0.4183
C18:1n7 (vaccenic acid)	1.76	1.59	1.53	1.86	2.09	1.62	1.41	0.374	0.8932	0.7747	0.4956
C18:2n6 (linoleic acid)	29.00	28.75	28.50	28.02	27.89	25.54	28.57	2.184	0.5721	0.8153	0.7508
C18:3n6 (r-linolenic acid)	0.05	0.05	0.05	0.06	0.08	0.06	0.06	0.019	0.4411	0.8684	0.8696
C18:3n3 (a-linolenic acid)	5.41	4.69	5.72	4.90	6.08	5.69	6.22	0.938	0.2915	0.9498	0.7298
C20:1n9 (eicosenoic acid)	0.33	0.27	0.35	0.32	0.30	0.37	0.33	0.073	0.7747	0.5768	0.9954
C20:4n6 (arachidonic acid)	4.04	3.00	4.04	3.25	2.89	2.46	3.60	0.902	0.5863	0.8840	0.5949
C20:5n3 (eicosapentaenoic acid, EPA)	0.20	0.13	0.18	0.19	0.11	0.14	0.15	0.046	0.3878	0.5867	0.9465
C22:4n6 (adrenic acid)	0.17	0.11	0.19	0.12	0.15	0.11	0.17	0.041	0.9850	0.9100	0.3022
C22:6n3 (docosahexaenoic acid, DHA)	0.24	0.15	0.20	0.19	0.15	0.22	0.22	0.058	0.7407	0.4801	0.9666
SFA	36.83	36.23	37.26	36.24	37.67	36.45	37.13	0.907	0.5231	0.9611	0.4872
UFA	63.17	63.77	62.74	63.76	62.33	63.55	62.87	0.907	0.5231	0.9611	0.4872
MUFA	24.05	26.88	23.85	27.03	24.97	29.33	23.87	3.068	0.9602	0.9413	0.3899
PUFA	39.12	36.89	38.90	36.72	37.36	34.22	39.00	2.686	0.7886	0.9029	0.4778
MUFA/SFA	0.65	0.75	0.64	0.75	0.66	0.81	0.64	0.098	0.9274	0.9587	0.3528
PUFA/SFA	1.06	1.02	1.04	1.01	0.99	0.94	1.06	0.071	0.6533	0.8528	0.6184

^NS^ Non-significant. C, control (boiled at 72 °C for 40 min); SEM, standard error of means; SFA, saturated fatty acid; UFA, unsaturated fatty acid; MUFA, monounsaturated fatty acid; PUFA, polyunsaturated fatty acid; T, temperature; t, time. T*t, The interaction between cooking time and temperature.

## Data Availability

The original contributions presented in the study are included in the article, further inquiries can be directed to the corresponding author.

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
