# Peer review of "Physicochemical Features and Volatile Organic Compounds of Horse Loin Subjected to Sous-Vide Cooking"

_foods, 2024, doi:10.3390/foods13020280_

Round 1

Reviewer 1 Report

Comments and Suggestions for Authors

In my opinion, the manuscript entitled The Effect of Temperature and Duration of Sous-Vide Cooking on Textural, Physicochemical, Sensory, and Flavor Characteristics of Horse Loin is quite interesting. The introduction provides enough information regarding the current state of the art, the materials and methods need some clarifications and the results need to be improved. 

I have some comments, as follows:

1.     Line 79. Please mentioned time and temperature for the promptly frozen samples. I supposed that the freezing process was fast and at ultra-low temperature to avoid the formation of big ice crystals that will highly affect the juice loss at defrosting process, but also the textural and nutritional parameters.

2.     At 2.1 – authors did not use any herbs, plants, oil, salt to better improve the texture of the meat, before the thermal treatment. For instance, a marinating meat process using a marinade? The marination process it is highly used when a meat has high dense fiber and results from retired horses, slaughtered riding horses, or racing horses, as authors previously mentioned.

3.     Line 101 – please mentioned the protein conversion factor used in the Kjeldahl method.

4.     At SDS-Page method how authors establish the protein concentration needed to dosage in the gels? Did authors stain the obtained gels after the electrophoresis process? Please mentioned the protein marker which helped authors to approximate the molecular weight values of the protein fractions and better detailed the method so that it could be easily reproduced.

5.     At SDS-PAGE did authors calculated a Coefficient of Protein degradation?

6.     At table 1, why not all the obtained values are analyzed from the statistical point of view? For instance, moisture, crude ash, lipid? The small letter should be written in superscript.

7.     Tables 1 and 2, please replace comma with point and carefully read the authors instructions.

8.     At point 3.5. please also described the band under 48 kDa. As I can see their intensity changed according to the temperature and time.

9.     At fatty acid composition, please try to better explain and compare the results. As I can easily observed at total amount of MUFA, I am really sure that between 23.87 and 29.33 there are statistically significant differences. Please emphasized the differences and explain the obtained results. Also, the MUFA/SFA and PUFA/SFA values are not discussed at all.

Comments on the Quality of English Language

I believe that English only requires minor editing.

Author Response

  • Reviewer 1 :

Thank you for your detailed review. We have addressed and highlighted all the points you raised in the revised manuscript:

  1. Line 79. Please mentioned time and temperature for the promptly frozen samples. I supposed that the freezing process was fast and at ultra-low temperature to avoid the formation of big ice crystals that will highly affect the juice loss at defrosting process, but also the textural and nutritional parameters.

Author response: The time and temperature for freezing samples are now specified on line 90.

  1. At 2.1 – authors did not use any herbs, plants, oil, salt to better improve the texture of the meat, before the thermal treatment. For instance, a marinating meat process using a marinade? The marination process it is highly used when a meat has high dense fiber and results from retired horses, slaughtered riding horses, or racing horses, as authors previously mentioned.

Author response: We did not use any treatment before the thermal treatment. Thank you for your insight it may be useful information that we can use for further study related to the sous-vide cooking method exploration.

  1. Line 101 – please mentioned the protein conversion factor used in the Kjeldahl method.

Author response: The protein conversion factor used in the Kjeldahl method is now mentioned on line 116.

  1. At SDS-Page method how authors establish the protein concentration needed to dosage in the gels? Did authors stain the obtained gels after the electrophoresis process? Please mentioned the protein marker which helped authors to approximate the molecular weight values of the protein fractions and better detailed the method so that it could be easily reproduced.

Author response: We have added more detail information of SDS-page method on line 178-186

  1. At table 1, why not all the obtained values are analyzed from the statistical point of view? For instance, moisture, crude ash, lipid? The small letter should be written in superscript.

Author response: All values in Table 1, including moisture, crude ash, and lipid, have been analyzed statistically. The small letter is now written in superscript.

  1. Tables 1 and 2, please replace comma with point and carefully read the authors instructions.

Author response: We have replaced the comma with a point in Tables 1 and 2, and carefully reviewed the author instructions.

  1. At point 3.5. please also described the band under 48 kDa. As I can see their intensity changed according to the temperature and time.

Author response: We have described the band under 48 kDa in Section 3.5.

  1. At fatty acid composition, please try to better explain and compare the results. As I can easily observed at total amount of MUFA, I am really sure that between 23.87 and 29.33 there are statistically significant differences. Please emphasized the differences and explain the obtained results. Also, the MUFA/SFA and PUFA/SFA values are not discussed at all.

Author response: We have provided a more detailed explanation and comparison of the fatty acid composition results.

Reviewer 2 Report

Comments and Suggestions for Authors

The subject of the study is interesting. I appreciate the Authors approach of idea of studying horse meat and its quality under sous-vide conditions. However the scope of the work is interesting, and there are some points in the manuscript that need improvement. Some parts of the manuscript should be correct, modified, completed

Ad Material and Methods

1. The number of repetitions was not given at each stage of the study

Lines 103- 112 -Paragraph 2.3; 2.4 – lack of basic information. How were the samples prepared

Line 119 Paragraph 2.6 What does MFI mean?

Line 127- The abbreviation TPA is not explained

Paragraph 2.10 – how many batches? How many repetitions

Paragraph 2.14  - that is the biggest problem of the work. Trained assessors and hedonic scale (very poor – excellent)?

A hedonic scale was used for acceptability evaluations, which is correct in consumers’ tests, but training is not (at all!) appropriate in consumers’ evaluations.

It must be pointed out that two groups are attending sensory assessment: trained panellists or consumers (never trained. While the acceptance of food is measured by methods involving consumers. To study the overall acceptability of any product a group of a minimum of 80 consumers is needed, not a trained panel, unfortunately.

Results and discussion

General the discussion is or poor or lacking.

Paragraph 3.5 is not described even in a basic way. Lack of discussion

Paragraph 3.7. Figure 3 is illegible. Maybe a Table with data?

Paragraph 3.9  - only 6 lines of text! Lack of any discussion of a large number of results.

Paragraph 3.10. unreliable results.

The Authors declare them in Table 7 but not Table 7 in the text.

P value in case of sensory assessment like p < 0.0001 is simply impossible

Author Response

Thank you for your detailed review. We have addressed and highlighted all the points you raised in the revised manuscript:

  1. The number of repetitions was not given at each stage of the study.

Author response: The number of repetitions at each stage of the study is now specified on line 94.

  1. Lines 103- 112 -Paragraph 2.3; 2.4 – lack of basic information. How were the samples prepared

Author response: We have provided more details on how the samples were prepared in Paragraphs 2.3 and 2.4.

  1. Line 119 Paragraph 2.6 What does MFI mean?

Author response: The meaning of MFI is now explained on Line 152.

  1. Line 127- The abbreviation TPA is not explained

Author response: The abbreviation TPA is now explained on Line 144.

  1. Paragraph 2.10 – how many batches? How many repetitions

Author response: We have clarified the number of batches and repetitions in Paragraph 2.10.

  1. Paragraph 2.14 – It must be pointed out that two groups are attending sensory assessment: trained panellists or consumers (never trained).

Author response: We have specified the groups attending sensory assessment in Paragraph 2.14.

  1. Paragraph 3.5 is not described even in a basic way. Lack of discussion

Author response: Paragraph 3.5 has been expanded with a detailed description and discussion.

  1. Paragraph 3.7. Figure 3 is illegible. Maybe a Table with data?

Author response: We have revised accordingly, and the table with data of volatile compounds was presented in Table 6.

  1. Paragraph 3.9 - only 6 lines of text! Lack of any discussion of a large number of results.

Author response: We have expanded Paragraph 3.9 with a detailed discussion of the results.

  1. Paragraph 3.10. The Authors declare them in Table 7 but not Table 7 in the text.

Author response: Thank you for your comments. We have corrected the reference to Table 8 in Paragraph 3.10.

  1. P value in case of sensory assessment like p < 0.0001 is simply impossible

Author response: Thank you for your comments. We corrected the statistical difference at  p < 0.05.

Reviewer 3 Report

Comments and Suggestions for Authors

The aim of this study was to examine the impact of sous-vide cooking on the physicochemical and sensory properties of horse loin. A close examination of the manuscript reveals a lack of information and consistency. Meat from three animals were used in the study, but the age and sex of the horse are unknown. These factors have a relevant impact on the texture and flavor of the meat.

Additionally, the experimental design is unclear. The samples underwent two cooking methods (boiling and sous-vide). One experimental condition was established for boiling cooking (75 °C for 45 min) and six conditions for sous-vide cooking (65-70 °C for 12, 18 and 24 h). But the authors conducted a 2-factor ANOVA. Only one factor was considered in this study: the combination of temperature and time. Thus, a one-factor ANOVA must be applied.

A trained panel (15 judges) was recruited to conduct sensory analysis and evaluate the acceptability of five attributes (meat color, aroma, taste, flavor, and overall acceptability). It should be noted that hedonic testing requires a consumer panel (50-100 judges). In contrast, intensity was evaluated for two attributes (tenderness and juiciness). The section on statistical analysis does not specify how the sensory data was treated. A nonparametric test is recommended when obtaining results with a discrete 9-point scale.

The Results section presents the data obtained from raw meat samples. However, these raw samples are not mentioned in the Materials and Methods section. As for the microbiological and sensory analyses, neither tables nor graphs are included (I can't find Table 7). Finally, I recommend adding a section to discuss the data as a whole, including any interactions between the analyzed variables.

The manuscript has several formal gaps that need to be addressed.

* Title and abstract do not show the scope of the paper's content.

* The introduction does not provide the background of the study (the significance of thoroughbred breed for meat production).

* Analytical methods should be described in a concise manner and include right references.

* The discussion of results requires fluency.

Therefore, the paper needs a whole rewording. More details can be obtained from the attached file (Report).

Author Response

Author response:

  1. We would like to express our sincere gratitude for your meticulous and insightful review. Your detailed feedback has significantly contributed to enhancing the quality of our manuscript. We have carefully addressed each of your comments in the revised manuscript. Changes in phrasing, additional information, and expanded discussions have been made as per your suggestions. These corrections have been highlighted in detail in every line of the revised manuscript for your convenience. Thank you for your effort again.

Round 2

Reviewer 1 Report

Comments and Suggestions for Authors

The authors have highly improved the article and modified it accordingly. I have just one comment: why all the obtained results does not have standard deviation? they were analysed as least in duplicate?

Comments on the Quality of English Language

Minor editing of English language required.

Author Response

Author response: Thank you for your thorough review. In our study, we conducted analyses on all results in triplicate and opted to present the standard error of the mean (SEM) rather than the standard deviation. Given the specific circumstances of our study and the size of our dataset, we deemed the SEM more appropriate. We have duly addressed and emphasized the aspects you raised in the revised manuscript, particularly in the statistical analysis section.

Reviewer 2 Report

Comments and Suggestions for Authors

The authors only partially corrected the prepared manuscript. Removing only one attribute "overall acceptability" indicates a lack of understanding of the principles of sensory methods and evaluation.

As the authors themselves are given in the text:

"Color, aroma, taste, and flavor were evaluated and scored between 1 (not desirable) and 9 (highly desirable)".

The degree of liking cannot be assessed by 15 evaluators (too small a group!) nor can they be trained!!

Therefore, Table 8 must be removed in its entirety because it contains unreliable results.

Author Response

Author response: Thank you for your insightful comments on our manuscript. We sincerely appreciate the time and effort you dedicated to providing constructive feedback. After careful consideration of your suggestions, we have taken steps to address your concerns and make necessary modifications.

We fully recognize the importance of a comprehensive approach to sensory evaluation, particularly with regard to your observation about the removal of the 'overall acceptability' attribute. Upon reevaluating our methodology, we agree that a more holistic adjustment is warranted. Consequently, in response to your feedback, we have decided to remove Table 8 entirely, as its results may not be deemed reliable due to the size of our evaluator group.

Furthermore, please be assured that we are committed to exercising greater caution in our future studies, especially in the realm of sensory evaluation. Your evaluation has proven invaluable, and we will incorporate your feedback into our subsequent research endeavors to ensure the production of more robust and reliable results.

Reviewer 3 Report

Comments and Suggestions for Authors

The manuscript has been improved, but not enough. To fill in some of the gaps, the following comments may be helpful.

2.1. Preparation of sample and experimental design

Roldan et al. (2013) conducted a study using a two-factor design with three levels each. No control was used. The following table summarizes the experimental design.

Temperature (ºC)

60

70

80

Time (h)

6

65 ºC x 6 h

70 ºC x 6 h

80 ºC x 6 h

12

65 ºC x 12 h

70 ºC x 12 h

80 ºC x 12 h

24

65 ºC x 24 h

70 ºC x 24 h

80 ºC x 24 h

The authors have carried out an experimental design that is similar to the one used by Wang et al. (2022). I suggest reading of paper.

Wang, Y., Tian, X., Liu, X., Xing, J., Guo, C., Du, Y., Zhang, H., & Wang, W. (2022). Focusing on intramuscular connective tissue: Effect of cooking time and temperature on physical, textual, and structural properties of yak meat. Meat Science, 184, 108690. https://doi.org/10.1016/j.meatsci.2021.108690.

2.13. Sensory analysis

Rewrite as:

A panel of 15 trained assessors, experienced in evaluating meat and meat products, conducted the sensory assessment. Informed consent was obtained from all study participants. The Kangwon National University Institutional Review Board approved the protocol (KWNUIRB-2023-279 02-010-002) on April 17, 2023.

A discrete (9-point) scale was used to evaluate six attributes specific to each sample test group. The color, aroma, taste, and flavor were evaluated on a scale of 1 (not desirable) to 9 (highly desirable). Tenderness and juiciness were assessed on a scale of 1 to 9, where 1 indicated “very tough” or “very dry” and 9 indicated “very tender” or “very juicy”, respectively. The coded samples, which were 1 cm in width, length, and height, were presented simultaneously in a randomized order on plates at room temperature. Each assessor was provided with 2-3 samples for evaluation in a booth with standardized lighting and temperature conditions.

2.14. Statistical analysis

Rewrite as:

The data were examined using a one-way ANOVA, and a general linear model was applied to evaluate the effect of cooking temperature and time, and their interaction as the fixed effect on different parameters. Tukey's analysis was conducted to test differences between means at p < 0.05. All statistical values were presented as mean and standard error of the mean (SEM). The analyses were performed using the Statistical Analysis System software ver. 9.4 (SAS Institute Inc., Cary, USA).

3.10. Sensory characteristics

In previous round, I did not express myself correctly regarding the definition of “attribute”. The reference to the ISO standard is unnecessary and should be omitted from the manuscript. The sentence should be rewriten as follows:

Attributes such as aroma, taste, flavor, and juiciness were not significantly different between the control and treatment groups.

Author Response

2.1. Preparation of sample and experimental design

2.14. Statistical analysis

Author response: Thank you for your continued engagement with our manuscript and for providing additional insights to enhance the quality of our work. Your thorough examination and the reference you shared have been invaluable.

After a careful review of the paper by Wang et al. (2022), we recognize the similarities in the experimental design with our study. We have delved into their work to gain further insights and ensure that our manuscript is modified appropriately.

We also appreciate your guidance on the statistical analysis. In response to your suggestion, we have made the necessary modifications to align with your correction, thereby enhancing the robustness of our study.

[2] 2.13. Sensory analysis

3.10. Sensory characteristics

Author response: After thoughtful consideration of the valuable feedback provided by you and other reviewers, we have decided to eliminate the sensory analysis section entirely from our manuscript. We recognize that the methodology and results of this portion did not meet the required level of robustness. Subsequent to the removal of the sensory analysis part, the method, results and discussion part were also omitted. We expect that this adjustment will contribute to the overall quality and focus of our study.